# Wi-CAS: A Contactless Method for Continuous Indoor Human Activity Sensing Using Wi-Fi Devices

**DOI:** 10.3390/s21248404

**Published:** 2021-12-16

**Authors:** Zhanjun Hao, Daiyang Zhang, Xiaochao Dang, Gaoyuan Liu, Yanhong Bai

**Affiliations:** College of Computer Science and Engineering, Northwest Normal University, Lanzhou 730070, China; haozhj@nwnu.edu.cn (Z.H.); dangxc@nwnu.edu.cn (X.D.); 2020211970@nwnu.edu.cn (G.L.); 2020211975@nwnu.edu.cn (Y.B.)

**Keywords:** Wi-Fi, device-free, human activity sensing, channel status information (CSI), hierarchical clustering, ensemble learning

## Abstract

With the new coronavirus raging around the world, home isolation has become an effective way to interrupt the spread of the virus. Effective monitoring of people in home isolation has also become a pressing issue. However, the large number of isolated people and the privatized isolated spaces pose challenges for traditional sensing techniques. Ubiquitous Wi-Fi offers new ideas for sensing people indoors. Advantages such as low cost, wide deployment, and high privacy make indoor human activity sensing technology based on Wi-Fi signals increasingly used. Therefore, this paper proposes a contactless indoor person continuous activity sensing method based on Wi-Fi signal Wi-CAS. The method allows for the sensing of continuous movements of home isolated persons. Wi-CAS designs an ensemble classification method based on Hierarchical Clustering (HEC) for the classification of different actions, which effectively improves the action classification accuracy while reducing the processing time. We have conducted extensive experimental evaluations in real home environments. By recording the activities of different people throughout the day, Wi-CAS is very sensitive to unusual activities of people and also has a combined activity recognition rate of 94.3%. The experimental results show that our proposed method provides a low-cost and highly robust solution for supervising the activities of home isolates.

## 1. Introduction

The outbreak of COVID-19 has had a huge impact on the world. At the time of writing, according to the World Health Organization (WHO), 213,725,662 cases of novel coronavirus infections have been confirmed worldwide, including 4,459,381 deaths [1]. In the face of this large-scale infectious disease, the World Health Organization pointed out in its guidance article that the isolation of confirmed persons and their close contacts can effectively cut off the transmission route of the virus [2]. Accordingly, a large number of residents in various countries are currently undergoing home isolation. Effective indoor activity monitoring of residents in isolation is needed to prevent accidents while they are in isolation. However, due to the large number of residents in isolation, it is not possible to monitor all residents in isolation even with a large number of additional volunteers, and the cost of manpower would be very high. Therefore, how to effectively and inexpensively monitor the indoor activities of these residents in isolation during isolation has become a problem that needs to be solved.

Traditional methods for human activity perception are mainly based on computer vision techniques [3] as well as sensor technologies [4]. However, these techniques have limitations. For example, computer vision-based detection uses an optical camera to record and identify the human body’s movements. Although this method can capture the subtle movements of the human body, the optical camera can only work in a well-lit environment, which limits its universality. Another sensor-based motion recognition system requires the user to carry additional sensor devices, which both impedes the user’s movement and violates the user’s privacy.

To compensate for the limitations of traditional technologies, human activity sensing using Wi-Fi signals has gradually gained attention. In recent years, related research based on Wi-Fi signals has also made great progress in action recognition [5], gesture recognition [6], fall detection [7], and person identification [8]. In previous studies, researchers have used Received Signal Strength (RSS) for human action recognition, such as the Wi-Gest [9] system to classify and recognize gestures by extracting the time-frequency domain features of RSS. Although good results have been achieved using RSS for human action recognition, RSS is susceptible to multipath effects resulting in unstable and noisy collected RSS signals, and these reasons can lead to erroneous detection. Fortunately, the Channel State Information (CSI) extracted from Wi-Fi signals using Orthogonal Frequency Division Multiplexing (OFDM) technology has a better performance. CSI is a more fine-grained metric that captures the wireless characteristics of the nearby environment and has better immunity to ambient noise. For example, the Wi-Act [10] system uses Extreme Learning Machine (ELM) for human motion classification. RDFID [11] can detect different motion patterns by extracting the frequency domain features of the motion using continuous wavelet transform. Compared with RSS, CSI has higher stability and greatly improves the accuracy of human motion detection.

Most of the current research on human activities has been conducted using neural networks or deep learning methods for accurate classification and recognition of continuous CSI action signals [12]. Although this method can achieve high recognition accuracy, the structure of deep learning is complex, which greatly increases the time consuming model training, and the computer hardware requirements for some model training are also high. Although the processing time is relatively short using traditional machine learning methods, the accuracy of action recognition is significantly reduced due to the huge amount of data of continuous activity signals. Therefore, how to balance the relationship between model processing time and accuracy of action recognition, i.e., to ensure high accuracy while reducing model processing time, has become a new challenge.

To solve the above problems, we propose Wi-CAS, an indoor human continuous activity sensing method based on Wi-Fi signals. Wi-CAS is divided into two phases: Off-line phase collects a large number of people’s daily movements (such as walking, jumping, sitting and standing, etc.) and abnormal movements (such as falling, etc.), and trains the HEC classifier after data processing. On-line phase, real-time activity data collection of home isolated persons on commercial Wi-Fi devices. After the filtering process, the Dynamic Time Warping (DTW) algorithm is used to segment the signal of the action occurring part into a signal image and then extract the Tamura texture features [13] of the action signal image. Finally, the feature dataset is fed into the trained HEC classifier to classify and recognize the active data. The main contributions of this paper are as follows.

We propose the Wi-CAS method, this method can use the CSI in the Wi-Fi signal to sense the long-term continuous activity of isolated people in the room. This method effectively avoids human contact, thus cutting off the transmission of the virus, and saves a lot of labor costs and facilitates the monitoring work of health care workers.We design an ensemble learning algorithm HEC based on hierarchical clustering. The method first reduces the similarity between base classifiers by hierarchical clustering, and then selects the appropriate number of optimal classifiers. This method effectively reduces system latency while improving system recognition accuracy.In this paper, we collect activity data of different people in a real home environment and form an open source database, and form a graph of the activity status of people in different time periods. This provides data support for health care professionals to analyze the status of isolated individuals. The experimental results show that Wi-CAS achieves a comprehensive activity recognition rate of 94.3%, and has high accuracy and robustness for abnormal movements.

The rest of this paper is organized as follows. Section 2 mainly summarizes the existing action recognition techniques. Section 3 describes the relevant principles and research methods. Section 4 specifies the implementation process of the Wi-CAS method. The performance of the Wi-CAS method is discussed and evaluated in Section 5. Finally, the work in this paper is summarized and conclusions are drawn in Section 6.

## 2. Related Work

This section presents current research work on human action recognition from both bound and unbound perspectives, and analyzes the advantages and disadvantages of each research work.

### 2.1. Bounded Activity Recognition

Bounded human motion recognition typically uses wearable sensors that are deployed in clothing, watches, etc. The sensors typically used are acceleration sensors, orientation sensors, gyroscopes, etc. Konstantinos Kyritsis et al. [14] detected wrist movements by obtaining acceleration and directional velocity in a smartwatch, and then combined a one-dimensional Convolutional Neural Network (CNN) with a Long and Short-Term Memory (LSTM) network to analyze the user’s eating patterns. Liu et al. [15] also integrated accelerometers, gyroscopes, and magnetometers as sensor nodes, collected human joint data and fused them, and used the fused data to detect a person’s driving movements. Akram Bayat et al. [16] used acceleration data generated by a user’s cell phone to identify certain types of human physical activity, and the paper designed a new digital low-pass filter to isolate the gravity acceleration component and body acceleration component in the original data. The high-frequency and low-frequency components of the data were also considered, and they selected five classifiers and used the probability mean of the five classifiers as the final classification result with an overall accuracy of 91.15%.

In addition, MZUA et al. [17] proposed a body sensor-based behavior recognition system. The article performs data fusion from multiple body sensors, such as Electrocardiogram (ECG), accelerometer, and magnetometer, and further enhances the extracted features by Kernel Principal Component Analysis (KPCA), and then Recurrent Neural Network (RNN) for behavior recognition. Song-Mi Lee et al. [18] collected three types of human activity data from users, namely walking, running, and stationary, through smartphone acceleration sensors, using a CN approach to recognize human activities and achieved 92.71% accuracy. Cuong Pham et al. [19] proposed SensCapsNet, a capsule network-based human activity recognition method for wearable sensors. SensCapsNet designs sensor network architecture for spatio-temporal data of wearable sensors. The experimental results show that the proposed SensCapsNet outperforms CNN and LSTM methods and achieves better results. Although bounded activity recognition has a greater advantage in recognition accuracy, it causes additional physical burden to users and affects people’s life, and it is expensive to deploy and maintain.

### 2.2. Unbound Activity Recognition

Unbound activity recognition is divided into four main categories: optical cameras, radar, Radio Frequency Identification (RFID) technology, and Wi-Fi computer vision-based human action recognition is used to analyze relevant actions by analyzing videos taken by optical cameras. Ahmad Jalal et al. [20] proposed that key action features can be extracted from sequences of continuous depth maps by depth imaging techniques. Classification recognition of actions using Hidden Markov Models (HMM). Radar-based human action recognition uses the signals emitted and received from radar devices and processes them for action recognition. Peng Lei et al. [21] proposed to use millimeter wave radar combined with CNN networks for human action recognition, which improves the recognition speed of the model and maintains high recognition accuracy through parallel processing. A multi-channel 3D neural network is proposed in the paper, which effectively improves the accuracy of recognition. Although the radar-based human action recognition has high accuracy, the processing of radar signal is more complicated and computationally intensive when performing accurate recognition. RFID-based human action recognition is based on the signal received by RFID reading tags, and then the signal is processed for action recognition. Lina Yao et al. [22] proposed to use RFID signals to recognize the daily activities of elderly people, and to deal with noisy and unstable RFID signals, they developed a compressed-aware, dictionary-based method that can use unsupervised subspace decomposition to learn a compact and information-rich set of activity dictionaries with a high action recognition rate. However, the RFID recognition method requires line-of-sight for action recognition.

The development and popularity of commercial Wi-Fi devices provides a solution to the shortcomings of the above mentioned techniques. Researchers have also proposed device-independent human motion recognition. This technology allows the user to recognize human motion without wearing any additional hardware devices, using only commercial Wi-Fi devices. The CSI extracted from Wi-Fi signals is physical layer information with finer-grained properties and has higher stability. It has also received a lot of attention from researchers in recent years. The SEARE [23] system uses CSI for green motion sensing to provide energy-efficient health management for users during exercise. The article uses dynamic time regularization to classify the action, detects the start and end of activity by First-Order Difference (FOD) and Fast Fourier Transform (FFT), and goes to sleep if the system does not detect motion within two minutes to achieve energy saving of the system. Hao Wang et al. [24] proposed the RT-Fall system, which uses CSI for human fall detection to achieve real-time automatic segmentation and detection of fall action signals. Sleppy [25] system uses a Gaussian Mixed Model (GMM) -based foreground extraction method to distinguish motion from resting state to monitor human sleep. Niu X et al. [26] proposed the WiMonitor system to monitor the indoor activities of elderly people living alone using CSI, and achieved good activity monitoring results. WiAnti [27] system proposes two adaptive subcarrier selection algorithms WiAntiPearson and WiAnti-DTW to solve the Co-Channel Interference (CCI) problem of CSI. Experiments demonstrate that both methods effectively reduce the impact of CCI and improve the performance of action recognition. These prior studies provide good theoretical support for the text’s approach.

## 3. Preliminaries

This section introduces the principles related to Wi-Fi sensing of human movement and the specific process of the Wi-CAS method.

### 3.1. Orthogonal Frequency Division Multiplexing

Orthogonal Frequency Division Multiplexing (OFDM) is a technique used to reduce the effects of frequency-selective and time-selective fading, and OFDM is also a multi-carrier technique that allows a single high-rate data stream to be transmitted over a large number of lower-rate subcarriers. According to OFDM, the transmit signal S(f) in the frequency domain is multiplied by the channel H(f), so the received signal R(f) satisfies.
(1)R(f)=S(f)·H(f)
where f denotes the subcarrier. Figure 1 shows the OFDM process.

As can be seen from Figure 1, OFDM uses multi-carrier modulation to modulate the signal into multiple sub-carriers, and information can be transmitted over multiple carriers simultaneously [28]. Therefore, OFDM can be used to send more information in parallel with higher frequency band utilization. Introducing cyclic prefixes as protection intervals during OFDM propagation. In the case where the protection intervals is greater than the maximum multipath delay extension, not only can the inter-symbol interference brought by multipath be maximized but also the inter-channel interference brought by multipath can be avoided. These advantages make OFDM an important technical basis for wireless sensing of Wi-Fi signals.

At the transmitter side, the digital signal carried by each subcarrier is superimposed on all subcarrier transmit signals by Inverse Fast Fourier Transform (IFFT). The real and imaginary parts of the signal are then changed to analog using Digital-to-Analog Conversion (DAC) and propagated into the air from the transmitter. After receiving these signals, the receiver performs Analog-to-Digital Conversion (ADC) of the signals. Then the FFT is performed to convert the signal to the frequency domain again. The CSI is eventually obtained in the frequency domain or in the form of multiple subcarriers, where each CSI portrays the amplitude and phase of a subcarrier.

So, CSI describes how the signal propagates in the channel, combining various effects such as time delay, amplitude fading and phase shift, describing the amplitude and phase in the frequency domain space corresponding to each subcarrier [29]. Since CSI is fine-grained physical information, subcarriers decoded from under the OFDM system. Thus CSI is more sensitive to the environment. Based on these characteristics of CSI, this paper uses CSI to sense the daily movements of isolated personnel.

### 3.2. Wi-Fi Sensing Model

When Wi-Fi signals are propagated in indoor spaces, they are affected by objects such as walls, ceilings, floors, tables and chairs, causing reflection, scattering and refraction of signals. These reflected and scattered signals will reach the receiver later in time compared to the signals in the direct path. Therefore, the final signal received by the receiver is the superposition of all path signals, which is the multipath effect [30]. According to the above principle, when the Wi-Fi signal touches the human body in the process of propagation, reflection, scattering and refraction will also occur, as shown in Figure 2.

Different actions cause different effects on the Wi-Fi signal, so different actions of the human body can be sensed by analyzing the changes in the signal received at the receiver. In Wi-Fi signals, the CSI provided by the physical layer describes the indicators of signal propagation in a multipath environment, which captures the multipath variation of the signal propagation path.

In the IEEE 802.11n protocol, the CSI can be extracted from all carrier bands of each channel in the Wi-Fi signal using OFDM technology. In this paper, we use Intel 5300 NIC, so we can get CSI information with 30 subcarriers by using CSI Tool and OFDM technology. Thus, under the IEEE 802.11n wireless protocol, CSI can be modeled as [31]:(2)$$\vec{Y}=H\vec{X}+Noise$$
where $\vec{Y}$ is the signal vector at the receiver, $\vec{X}$ is the signal vector at the transmitter, and Noise is Gaussian white noise. H is the channel matrix, which can be estimated as CSI, as shown by:(3)$$H(f)=[H(f_{1}),H(f_{2}),\cdots ,H(f_{n})],n\in [1,N]$$
where $H(f_{n})$ is the CSI on the nth subcarrier, N is the number of subcarriers, and in this paper N=30. Both amplitude and phase are included in $H(f_{n})$. The equations are:(4)$$H(f_{n})=||H(f_{n})||\cdot e^{j\sin\theta_{n}}$$
where $\left||H(f_{n})|\right|$ is the amplitude of the nth subcarrier and $e^{j\sin\theta_{n}}$ is the phase information. When the Wi-Fi signal passes through the human body, the CSI amplitude will change accordingly, and different movements of the human body will trigger different amplitude changes. Based on this principle, this paper uses CSI amplitude to sense the human body’s activities.

### 3.3. Overview

#### 3.3.1. Research Objectives

The Wi-CAS method in this paper takes the daily activities of isolated persons as the research target, which can identify the activities of isolated persons and determine the abnormal activities. Finally, the activity status map of the isolated person in any time range can be generated based on the collected activity data, in order to facilitate medical personnel to analyze the physical condition of the isolated person through the activity status map. The first step is to collect CSI data on commercial Wi-Fi devices for everyday actions such as walking, sitting, standing up, lying down, squatting, falling, etc.

These six actions are selected because they are the starting actions of an activity state. For example, after detecting the user’s sitting down action, the user’s activity state can be judged as sitting activity before detecting the rising action, and after detecting the lying down action, the user can be judged to be about to enter the sleep state. When a fall is detected, the user’s abnormal movement can be determined and the user and the medical personnel can be alerted for timely treatment. By detecting different movements, the user’s different activity states can be determined.

In order to collect CSI data for daily movements, we use the Multiple Input Multiple Output (MIMO) system of 1×3 [32]. One antenna is set up on the transmitter side of the commercial Wi-Fi device and three antennas are set up on the receiver side, for a total of three channels. Since each channel carries the information of the action, we uses the action data of all three channels. Figure 3a shows the CSI data of user activity collected for one channel over a period of time. We refer to the method in the literature [25] to relate the user’s amplitude information to the activity status. Activity status below 0.5 is stationary or sleeping, 0.5–1 is sitting activity, 1–1.5 is standing activity, and above 1.5 is vigorous exercise. Figure 3b shows a graph of the user activity status during the period based on the data in Figure 3a. The blue part of the figure shows the stationary state, the green part shows the standing activity, the red part shows the abnormal activity, and the yellow part shows the sitting activity.

As can be seen in Figure 3a, there is a more significant difference in CSI amplitude between the standing and sitting. After converting the amplitude to the active status, it can be seen that the active state value for the sitting is lower than that for the standing. Hence, we can record the daily activities of isolated people more accurately and visually.

#### 3.3.2. Method Flow

In this section, the process of Wi-CAS method is elaborated. The Wi-CAS method is divided into four phases, which are data collection and processing, feature selection, offline training phase and online recognition phase. The flow of Wi-CAS is shown in Figure 4.

The CSI action data for each daily action is first collected on a commercial Wi-Fi device. In the data processing stage, to remove the interference of noise, we first use median filtering to remove outliers and then we uses Butterworth low-pass filter to remove ambient noise. After that, wavelet transform is used to smooth the data of each daily action. In the feature extraction stage, if the input data is off-line action data, the texture feature is directly extracted; if it is online activity data, the action segmentation is performed first and then the feature extraction is performed. In the off-line phase, three classifiers are constructed as base classifiers, and then the three classifiers are clustered using hierarchical clustering, and the best classifier in each cluster is selected according to the AUC and then integrated to form the final classifier and trained with the collected samples. After collecting the data in the online stage, the data are processed identically and fed into the classifier to output the final action classification results.

## 4. Methodology

This section specifies the various aspects of the Wi-CAS method, including data noise reduction and smoothing, feature extraction, and the construction of the HEC classifier.

### 4.1. Motion Data Processing

The acquired raw data contains a large amount of noise, which is on the one hand due to the multipath effect in the environment and other interferers on the signal. On the other hand, it is because of the Carrier Frequency Offset (CFO) as well as Sampling Frequency Offset (SFO) of the device itself [33]. Therefore, we first use the method in the literature [34] to eliminate the CFO as well as the SFO. Most of the current studies use a single filter for ambient noise. Although the use of a single filter has good performance against environmental noise, it is found through experiments that the single filter filtering effect is not good for some outliers. To solve this problem, we first uses median filtering to remove outliers, and then uses Butterworth low-pass filter for those Gaussian noises that are not well handled by median filtering, and finally uses wavelet transform for smoothing. Figure 5 shows the process of data processing.

From Figure 5a, it can be seen that the collected CSI action raw data contains a large number of outliers and environmental noise, which mask the real action signal. In order to remove the outliers, we first uses median filtering to filter the action data. Although median filtering has good performance in removing outliers, it performs poorly for ambient noise. Therefore, to address this point, we then uses the Butterworth low-pass filter to reduce the noise of the action signal again, and the action signal is effectively removed from the environmental noise after filtering by two filters. After noise reduction, the ambient noise in the action signal is effectively removed. The noise effect is reduced, but the action characteristics are still not obvious. In order to make the action features more prominent, we use wavelet transform to further process the action signal. In this paper, Symlets wavelet is selected as the wave base function, and the wavelet system is selected as sym8 wavelet; sym8 wavelet has a support range of 2N−1, vanishing moment of N, good regularity and symmetry, and can reduce the distortion rate of the wavelet transform when analyzing and reconstructing CSI information. The raw CSI motion signal after processing is shown in Figure 5b.From the processing results, we can see that the action signal becomes smoother, and its action characteristics are more obvious after using wavelet transform processing, and the noise interference is further eliminated.

After the action signal is smoothed, in order to further reduce the computational complexity, the optimal subcarrier of the action is extracted by using Principal Component Analysis (PCA) in this paper [23]. The action data processed by PCA algorithm is shown in Figure 5c. From the figure, it can be seen that the action information is retained to the maximum extent and the computational complexity is effectively reduced after processing with PCA algorithm.

### 4.2. Motion Segmentation

In order to accurately identify the activities of isolated personnel, action segmentation of the collected real-time data is required to segment the continuous activity signals into individual action signals. In this paper, we first use the Dynamic Time Warping (DTW) algorithm to match continuous activity signals with individual action signals [35]. The DTW algorithm, proposed by Japanese scholar Itakura in the 1960s, is a method to measure the similarity of two time series of different lengths and is mainly used for pattern matching. In DTW, the comparison is performed by allowing a slight distortion of the time-series signals, using a dynamic programming-like approach to match two time-series signals with a distortion [36]. Using the DTW algorithm, it is possible to match a single action in a continuous active signal, and then the successfully matched action signal can be segmented. Figure 6a shows the process of action signal matching using the DTW algorithm.

In this paper, the input signals are matched using the open source DTW packet that comes with the MATLAB 2018a program. After matching, all the individual action signals in the continuous activity can be matched, and after that the continuous activity signals can be matched by continuously using the DTW algorithm. The matching results are shown in Figure 6b. Finally, the action signal with successful matching is intercepted to realize the segmentation of the action signal, and the segmentation result is shown in Figure 7.

By the above method, the continuous action signal can be segmented, and the segmented action signal can further improve the recognition efficiency. It is worth mentioning that although the action signals can also be identified using the DTW algorithm. However, this approach requires tagging all the data collected offline, increasing the computational overhead of the system. It is proved through experiments that the results of action recognition using only DTW are low, so we use only DTW for action signal matching.

### 4.3. Feature Extraction

Compared with signal data, image data has more stable feature information [37], and the texture features of the image data portray the recurring local patterns in the image with their alignment rules, with good noise immunity and rotation invariance [38]. In order to make the recognition of action more accurate, in this paper, after segmenting the continuous activity, the segmented action signal is processed as a signal image and then its texture features are extracted. The Tamura texture feature proposed by Tamura et al. was shown to have better performance in image classification [39], so we extracted the Tamura texture feature of each signal image as the feature of this action.

The six components of Tamura texture features are coarseness, contrast, directionality, linearity, regularity and roughness, and according to the literature [30], the first three components are particularly important. Therefore, in order to minimize the computational complexity while ensuring the recognition accuracy, we selected the first four components as the feature values of the action.

Roughness is a quantity that reflects the particle size in the texture and is the most basic texture feature. Its calculation formula is as follows.
(5)$$F_{\mathrm{crs}}=\frac{1}{m\times n}\sum_{m}^{i=1}\sum_{n}^{j=1}S_{best}(i,j)$$
(6)$$S_{best}(i,j)=2^{k}\;(k=0,1,\ldots ,5)$$
where i, j are the number of columns and rows at the location of pixel points. $S_{best}(i,j)$ is the pixel location for the best size of the CSI data image. k is a constant to set $S_{best}(i,j)$. m, n are the number of rows and columns of pixel points in the CSI data image. Contrast is obtained from the statistics of the pixel intensity distribution, which is calculated as:(7)$$F_{con}=\frac{\varepsilon }{(\alpha_{4}{)}^{n}}$$
(8)$$\alpha_{4}=\frac{\varphi_{4}}{\varepsilon^{4}}$$
where $\varphi_{4}$ is the fourth moment, $\varepsilon^{2}$ is the variance, and n generally takes the value 0.25. Contrast gives a global metric for the entire signal image. Directionality is a global characteristic of a given region, describing how the texture diverges or concentrates along certain directions. The overall directionality of the CSI data image can be obtained by calculating the sharpness of the peaks in the histogram at:(9)$$F_{\mathrm{dir}}={\sum_{n_{k}}^{k}\sum_{\theta \in W_{k}}\left(\theta -\theta_{k}\right)}^{2}H_{d}(\theta )$$
where k is the peak of the histogram, $n_{k}$ is the peak of all histograms, $W_{k}$ represents all the discrete regions contained in that peak, $\theta_{k}$ is the center of the wave, and $H_{d}$ is the histogram. Linearity is a case of texture consisting of line segments. Its calculation formula is shown as follows.
(10)$$F_{lin}=\frac{\sum_{j}^{n}\sum_{j}^{n}P_{Dd}(i,j)\cos[(i-j)\frac{2\pi }{n}]}{\sum_{j}^{n}\sum_{j}^{n}P_{Dd}(i,j)}$$

According to the description of literature [30], it is necessary to build the directional co-occurrence matrix first, and $P_{Dd}$ is the local directional co-occurrence matrix of size n×n in the distance. $P_{Dd}(i,j)$ is a matrix element representing the frequency of occurrence of two adjacent pixels that are d apart along the edge direction in the image, where one direction is coded as i and the other’s direction is coded as j. Since the data of three channels are used in this paper, a total of Tamura texture features of three channels of this action need to be extracted. After extracting the coarseness, contrast, directionality, and linelikeness of each image, all the data obtained are stitched to form a feature matrix of 4×3.

### 4.4. Construction of HEC

The main idea of the ensemble classification method is to combine multiple classifier models together into a new composite classifier and with better classification results compared to a single classifier [40]. Each training of this algorithm generates a new classifier that will be added to the whole [41].

The key to ensemble learning reducing the error is the diversity and accuracy of the base classifiers among them, and the higher the diversity and accuracy of the base classifier the lower the error of the output results. So, in order to reduce the similarity between base classifiers, we introduce the idea of clustering. The principle is to use clustering to gather base classifiers with high similarity together and use the AUC index as the evaluation criterion. Since the larger the AUC index, the better the classification effect of the classifier, so in order to improve the accuracy of the base classifier, the classifier with the largest AUC index in each cluster is finally selected as the final base classifier. So, in order to obtain more accurate action recognition results, we build an integrated classification method based on hierarchical clustering based on the advantages of integrated learning and hierarchical clustering.

A total of 2400 sets of action data are collected and processed to form a feature matrix. All feature matrices are pooled to form the action feature dataset, and the feature dataset is divided into two parts, two-thirds as the training set and one-third as the test set. In the training set, some samples are randomly extracted each time to build Classification and Regression decision Tree (CART), logistic regression and naive Bayesian classifier.

The construction of the CART decision tree requires the calculation of the Gini index of the sample, which indicates the purity of the sample set, and the smaller the Gini index, the higher the purity of the sample. The CART decision tree is constructed by calculating the Gini index of all feature matrices in the feature dataset that can be used to classify the sample attributes. The Gini index is calculated as follows.
(11)$$Gini(p)=\sum_{n=1}^{N}p_{n}(1-p_{n})=1-\sum_{n=1}^{N}p_{n}^{2}$$
where N is the number of samples and $p_{n}$ is the probability that the sample belongs to category n. So, for a given sample set M, the Gini index is:(12)$$Gini(M)=1-\sum_{n=1}^{N}(\frac{\left|C_{n}\right|}{\left|M\right|}{)}^{2}$$
where $C_{n}$ is the nth sample value. At this point, for this sample set M, M can be divided into two parts $M_{1}$ and $M_{2}$ according to whether its feature A takes a certain value a. Conditional on A, the Gini index of M is expressed as the following equation.
(13)$$Gini(M,A)=(\frac{\left|M_{1}\right|}{\left|M\right|})Gini(M_{1})+(\frac{\left|M_{2}\right|}{\left|M\right|})Gini(M_{2})$$

Gini(M,A) represents the uncertainty of feature A after taking a values to split the sample set M. It is equivalent to finding the Gini index for each of the two sets divided according to A and then obtaining the Gini index for M under feature A according to the empirical probability expectation. The significance of the Gini index is used to indicate uncertainty; the larger the Gini(M), the higher the uncertainty. The higher the Gini index of the sample the better the result of the generated decision tree. Using the above formula to calculate the Gini index for each input sample, a decision tree can be formed and the different actions can be classified.

The logistic regression model is usually represented by a sigmoid function as follows.
(14)$$g(z)=\frac{1}{1+e^{-z}}$$
where z is the variable of the function. In order to judge how good or bad the predictive ability of this function is, a loss function needs to be introduced to represent how good or bad the logistic regression function $h_{\theta }(X)$ is. Therefore, for a single sample the loss function is:(15)$$\mathrm{Cos}t(h_{\theta }(X),y)=\{\begin{array}{ll} -\log(h_{\theta }(X)) & y=1\\ -\log(1-h_{\theta }(X)) & y=0 \end{array}$$
where y=1 indicates that the sample is positive and y=0 indicates that the sample is negative. For all samples in the training set, the mean value J(θ) of the jointly caused loss function is:(16)$$J(\theta )=-\frac{1}{m}\sum_{i=1}^{m}\mathrm{Cos}t(h_{\theta }(X^{i}),y^{i})$$
where m is the sample size. Bringing the loss function Cost into the above equation yields.
(17)$$J(\theta )=-\frac{1}{m}\sum_{i=1}^{m}\left[\left(y^{i}\right)\log\left(h_{\theta }(X^{i})\right)+\left(1-y^{i}\right)\log\left(1-h_{\theta }(X^{i})\right)\right]$$

For the prediction probability function $h_{\theta }(X)$, the probability value of predicting that the sample is positive ranges from 0 to 1. A logistic regression classifier can be used to classify different actions by calculating the probability density of each input feature set.

The process of constructing a plain Bayesian classifier relies on the following equation:(18)$$y_{k}=\arg\;\max\left(P(y_{k}|x)\right)\;y_{k}\in y$$
(19)$$P(y_{k}|x)=\frac{P(x|y_{k})P(y_{k})}{P(x)}$$

Suppose the input dataset $x=(x_{1},x_{2},\cdots ,x_{D})$ represents a data object containing D-dimensional attributes. The training set S contains k categories, denoted as $y=(y_{1},y_{2},\cdots ,y_{k})$. The data object x to be classified is known, and the category to which x is predicted to belong, and $y_{k}$ is the category to which x belongs. According to Bayes’ theorem, $P(y_{k}|x)$ is calculated as shown in Equation (12). P(x) is equivalent to a constant for $P(y_{k}|x)$. Therefore, if you want to get the maximum value of $P(y_{k}|x)$, you only need to calculate the maximum value of $P(x|y_{k})P(y_{k})$. If the prior probability $P(y_{k})$ of the categories is unknown, these categories are usually assumed to be equally probable, i.e., $P(y_{1})=P(y_{2})=\cdots =P(y_{k})$.

Assuming that the attributes of data object x are independent of each other, $P(x|y_{k})$ is calculated as follows.
(20)$$P(x|y_{k})=\prod_{d=1}^{D}P(x_{d}|y_{k})=P(x_{1}|y_{k})P(x_{2}|y_{k})\cdots P(x_{D}|y_{k})$$

As our attributes $A_{d}$ are continuous attributes, they all obey a Gaussian distribution with mean and standard deviation, therefore, $P(x_{d}|y_{k})$ is calculated as:(21)$$P(x_{d}|y_{k})=G(x_{d},\mu_{y_{k}},\varphi_{y_{k}})$$
where $\mu_{y_{k}},\varphi_{y_{k}}$ are the mean and standard deviation of data objects belonging to category $y_{k}$ in the training set under attribute $A_{d}$. Based on the above formula, a Naive Bayesian classifier can be constructed to classify the action feature set.

After that, all the generated classifiers are hierarchically clustered to select the classifier that performs better for action classification. The generalized diversity of each classifier is first calculated as the feature of that classifier. The generalized diversity represents the similarity between two classifiers. So, the generalized diversity R of a single classifier is calculated by the following formula:(22)$$R=\sum_{i=1}^{L}\frac{i}{L}p(i)$$
where p(i) denotes the chance that i randomly selected classifiers misclassify on a randomly selected sample. L is the number of all classifiers. We use R as a feature of this classifier to perform hierarchical clustering on all classifiers.

Hierarchical clustering uses Euclidean distance to calculate the similarity between data points of different categories. Its formula is:(23)$$Dis=\sqrt{{\left(x_{1}-y_{1}\right)}^{2}+{\left(x_{2}-y_{2}\right)}^{2}}$$

If the features of a classifier are considered as a data point, the distance value between two data points can be calculated according to the formula of Euclidean distance by using the following formula:(24)$$Dis=\sqrt{{\left(R_{1}-R_{2}\right)}^{2}}$$
where, $R_{1}$, $R_{2}$ are the feature data points of two classifiers. The partial classifier hierarchical clustering process is shown in Figure 8.

Figure 8 shows the process of hierarchical clustering for some of the classifiers, from which it can be seen that the two classifiers with the closest Euclidean distance are clustered into a single cluster. After that, the two clusters with the closest Euclidean distance will be clustered into one large cluster until the number of clusters becomes 1. The advantage of hierarchical clustering is that the number of clusters generated by clustering can be controlled so that the optimal number of classifiers can be selected.

The AUC index is an indicator of the classifier’s classification effectiveness, and the higher the AUC index, the better the classifier’s effectiveness. The AUC index is calculated as follows.
(25)$$AUC=P(P_{+}>P_{-})$$
where $P_{+}$ is the probability of positive samples and $P_{-}$ is the probability of negative samples. The optimal base classifier selected by the above operation achieves both high accuracy of individual classifiers and reduces the correlation between individual classifiers. The pseudo-code for selecting the optimal classifier is shown in Algorithm 1.
**Algorithm 1.** Selection of The Optimal Classifier**Input:** Generalized diversity $R=[R_{1},R_{2},\ldots \ldots R_{m}]$, Distance metric function Dis, Cluster number K
**Output:** Optimal classifier $C=\{C_{1},C_{2},\ldots ,C_{n}\}$
1: ω=m //Set the Optimal number of current clusters2: **while** (ω>K) **do**3: Find the nearest two clusters $H_{i*}$, $H_{j*}$ //i, j are the number of rows and columns of the matrix respectively.4:  $H_{i*}=H_{i*}\cup H_{j*}$ //merge $H_{i*}$ and $H_{j*}$
5:  **for** $j=j^{*}+1,j^{*}+2,\ldots ,\omega$ **do**6:  $H_{j}\to H_{j-1}$ //Renumber $H_{j}$ as $H_{j-1}$
7:  **end for**8:  Delete row $j^{*}$ and column $j^{*}$ of the distance matrix9:  **for** j=1,2,…,ω−1 **do**10:  $O(i^{*},j)=Dis(H_{i*},H_{j})$; //$Dis(H_{i*},H_{j})$ is the distance matrix.11:  $O(j,i^{*})=M(i^{*},j)$
12:  **end for**13: ω=ω−1
14: **end while**15: **for** α=1,2,…,m **do**16:  **for** n=1,2,…,K **do**17:  $AUC_{\alpha }=P(P_{+}>P_{-})$ //Calculate the AUC index for each classifier18:  $C_{n}=\max(AUC_{\alpha })$
19:  **end for**20: **end for**

First, set the number of current clusters ω, then find the two closest clusters $H_{i*}$ and $H_{j*}$ and merge them, and renumber the new clusters as $H_{j-1}$. Delete row $j^{*}$ and column $j^{*}$ of the distance matrix. The above steps are iteratively combined until ω clusters are produced. Finally, the AUC indices of all classifiers in each cluster are calculated, and the classifier with the highest AUC index in each cluster is selected to form the final set of classifiers $C_{n}$. It is concluded from the subsequent experiments that the classification result is optimal when the number of classifiers is 600, so this paper generates 3600 base classifiers for hierarchical clustering and sets the number of generated clusters to 600. The HEC model is obtained by integrating all classifiers in $C_{n}$. Finally we use the generated HEC model to identify the daily activities of isolated persons.

## 5. Experimental Evaluation

### 5.1. Experimental Setup

The experimental equipment in this paper is two Thinkpad x201i laptops with Intel 5300 NIC chips. Both NIC chips support IEEE 802.11n protocol.

Both laptops can use CSI Tool [42] to extract the CSI information from the Wi-Fi signals. As both laptops are equipped with independent batteries, they can perform normal operation for a period of time even in a powerless environment, which is a good response to possible power outages. In addition, both laptops are installed with three external antennas, each with a signal gain of 6 dBi, and one transmitting antenna and three receiving antennas are set in this paper. The transmitter and receiver are connected through the monitor mode, and the transmitting band is set to 2.4 GHz, and the transmitting rate is set to 100 packets of data per second.

We arranged the equipment in a real home environment and verified it in four different scenarios, and the floor plan and realistic view of the experimental scenario are shown in Figure 9.

The four experimental scenarios are bedroom 1, bedroom 2, foyer and living room. The devices were placed in each of the four scenarios to validate the Wi-CAS. A total of 20 experimenters, 10 males and 10 females in the age group of 10 to 60 years old, were recruited and divided into four groups of five persons each according to gender and age. The experiment-related settings are shown in Table 1.

All experimenters collected action data in four scenarios, and each person collected 30 sets of data per action as samples, 600 samples per scenario. A total of 2400 samples were collected, of which 70% of the samples were used as the training set and 30% of the samples were used as the test set. Considering that in the real isolation environment, the positions of isolated people moving around the room are random, so in order to make the data in the sample set richer, the positions of the experimenter collecting data are not restricted during the experiment. So, the obtained action data all contain a Line-of-Sight (LOS) environment and a Non-Line-of-Sight (NLOS) environment.

### 5.2. Experimental

#### 5.2.1. Performance of Cross-Domain Action Recognition

To verify the cross-domain performance of Wi-CAS, we validate it in four different home environment domains. The equipment was arranged in room 1, room 2, living room, and hall shown in Figure 9, and the experimenters were allowed to do six actions in each of the four environments. Additionally, to verify the effect of LOS and NLOS on Wi-CAS, in this paper, we let experimenters perform six actions in any of the four environmental domains. The combined action recognition results of the four environmental domains are shown in Figure 10.

As shown in Figure 10, bedroom 1 and bedroom 2 are closer to each other in terms of action recognition results because of their similar room configurations. The hall is more open than the other environmental domains so the motion recognition results are the highest. The living room has more distractions and a complex environment, so the action recognition result is the lowest among the four environmental domains. In addition, because the experiment does not fix the location of the experimenter to do the action, so the experimental results also include the LOS and NLOS. From the experimental results, it can be seen that Wi-CAS performs better regardless of whether the action occurs in the LOS or NLOS. From the combined results, the average action recognition rate is above 90% in the four environmental domains, which proves that Wi-CAS has better recognition performance for cross-domain actions.

#### 5.2.2. Impact of Through-Wall Detection

To further validate the performance of Wi-CAS, the transmitter and receiver were set up at location 1 and location 2 marked in Figure 9, and the experimenter was asked to perform the corresponding activities in the above four scenarios to verify the performance of Wi-CAS in the presence of wall obstruction. The action recognition rate of each scene is shown in Figure 11.

Figure 11 shows the average action recognition rate of the experimenters in the four scenarios. As can be seen from the results in the figure, the activity recognition results are higher in bedroom 1 and the living room, which is due to the fact that the signal crosses fewer walls when it passes through these two environments, so the activity signal receives less interference. The activity recognition rate in bedroom 2 is the lowest, which is due to the fact that the signal traverses multiple walls when it is transmitted from the transmitter through bedroom 2 to the receiver, resulting in lower signal strength and lower action recognition rate. However, from the comprehensive practical results, the activity detection rate of Wi-CAS through walls for all four scenarios is above 85%, indicating that Wi-CAS can still show high performance when there are walls in the way.

#### 5.2.3. Impact of User Diversity

User diversity also has an impact on Wi-CAS, so this paper will explore the impact of user diversity in terms of both individual actions as well as periodic activities. It can be seen from the previous experiments that since the environments of bedroom 1 and bedroom 2 are similar, the action recognition results are not very different. Therefore, firstly, one experimenter is allowed to do six actions in three environments, and after that, three males and three females of different age groups are randomly selected to do a total of six daily actions in each of the three environments to verify the recognition rate of Wi-CAS for different people’s actions. The comprehensive experimental results are shown in Figure 12.

As can be seen in Figure 12a, Wi-CAS has a high recognition rate for different actions of the same person. In Figure 12b, experimenters 1–3 are males and the rest are females. From the experimental results, it can be seen that the action recognition results of males are slightly higher than those of females. This is due to the fact that the size of males is slightly larger than that of females, so the amplitude and range of the actions are larger, and the impact on the CSI is more obvious, so the action recognition rate of males is higher. In summary, the action recognition rates of different experimenters were above 90% in all three environments. It proves that Wi-CAS has high robustness to different users in terms of individual action recognition.

For the investigation of periodic activities, we allowed two experimenters to enter the home environment (shown in Figure 9) at different times to simulate isolated people for daily activities. Among them, the first experimenter simulates a normal isolated person to conduct the experiment, and the second experimenter simulates an unexpected situation such as falling down and leaving the isolated place privately to verify whether Wi-CAS can detect for unexpected situations. Each of the two experimenters were allowed to record their activities at different time periods, as shown in Table 2 and Table 3. Afterwards, the CSI information of their activities for 24 h was collected, and the information was processed by Wi-CAS to form an activity state diagram of the experimenter. The activity state diagram was compared with the activities recorded by the experimenter, and the comparison results are shown in Figure 13.

From the comparison results, it can be seen that the activity recorded by the first place experimenter basically matches with the activity state graph generated via the Wi-CAS. The second experimenter’s activity state diagram, with the abnormal state detected by Wi-CAS in red, is also largely consistent with what the experimenter recorded. For the Wi-CAS system, an alert to the health care worker is issued immediately after the abnormal action is detected. From the experimental results, it can be seen that for periodic activities, Wi-CAS can record accurately and has a high sensitivity to abnormalities in them. It proves that Wi-CAS shows better performance in both individual actions and periodic activities for different users.

#### 5.2.4. Impact of Different Platforms

Portability is also an important measure of system performance. So, we tested it on Intel 5300 NIC platform and Atheros NIC platform respectively. The Atheros NIC platform consists of two TP-Link routers equipped with Atheros AR9580 NICs. The motion data were collected from the two platforms separately for comparison after processing using Wi-CAS. Figure 14 shows the parameters and action recognition results of the two NIC platforms.

From the experimental results, it can be seen that Wi-CAS performs well on both NIC platforms, and it can be seen from the confusion matrix that Wi-CAS can also distinguish well for similar actions such as squatting and falling on the Atheros NIC platform. The experimental results show that this solution shows better recognition results on both Intel 5300 and Atheros AR9580 NIC platforms, which proves the strong portability of Wi-CAS.

### 5.3. Evaluation

#### 5.3.1. Impact of Base Classifier

The recognition accuracy of the ensemble classifier also depends on the performance of the individual base classifiers. Therefore, the performance of different base classifiers was first compared. We compared Decision Tree (DT), Naive Bayesian (NB), Logistic Regression (LR), k-Nearest Neighbor (KNN) classifier, Linear Discriminant Analysis (LDA) classifier, and General Bayesian Network (GBN) classifier. The comparison results are shown in Figure 15a.

From the comparison results, we can see that the accuracy of decision tree, naive Bayesian and logistic regression algorithms are above 85% and higher than the rest of the classification algorithms, so these three classifiers are chosen as the base classifiers of the integrated classifier in this paper. In addition, the number of base classifiers of the ensemble classifier also affects the recognition results, so we tested the effect of different numbers of base classifiers on the experimental results. The experimental results are shown in Figure 15b.

From the experimental results, it can be seen that the error rate gradually decreases when the number of base classifiers approaches 600. When the number of base classifiers is about 600, the error rate is about 0.065, and when the number of base classifiers exceeds 600, the error rate gradually stabilizes at about 0.064. However, more base classifiers will lead to higher computation time for classification, so when the number of base classifiers is 600, the system overhead can be reduced while ensuring optimal recognition results.

#### 5.3.2. Impact of Sample Size

Different sample sizes also have an impact on the recognition results, so we validated this. We compared the effect of different samples on the recognition results of the four algorithms. The comparison results are shown in Figure 16.

As can be seen from the figure, compared with other algorithms, HEC always maintains a high recognition accuracy as the number of samples increases. The recognition result of HEC is the highest when the number of samples hits 700, and after that, the recognition result gradually remains stable as the number of samples increases. Therefore, HEC shows the best performance when the number of samples hits 700. Besides, although the method using Bidirectional Recurrent Neural Network (Bi-RNN) also has high recognition accuracy, the method gradually performs unstable as the number of samples increases, so from this point of view HEC has higher recognition performance.

#### 5.3.3. Comparison of Different Algorithms

To verify the performance of the Wi-CAS system, we compares it with traditional classification algorithms and previous research works, respectively. Traditional classification algorithms are compared with Random Forest (RF), Dynamic Time Warping (DTW) algorithm and Bi-RNN in this paper, and ROC curves are used as evaluation criteria. The comparison results are shown in Figure 17.

From the comparison results, it can be seen that when the True Positive Rate (TPR) reaches 0.8, the False Positive Rate (FPR) of HEC is only 0.04, which has the best performance, and the FPR of Bi-RNN is 0.22 which is slightly lower than that of HEC. The FPR of DTW reaches 0.68, which has the worst performance. From the comparison results, we can see that HEC has the best performance, higher than Bi-RNN algorithm. DTW has the worst performance. Compared with the three algorithms, HEC has a better performance. In addition, we have validated all the algorithms proposed in the literature [23,24,25]. The comparison results are shown in Figure 17b.

The comparison results in Figure 17b show that the recognition result is 90.8% after processing the data using the SEARE [23] method and 93.6%. After processing with Sleepy [25] method, the recognition result was 93.6%. The identification result after processing the data using the RT-Fall [24] method was 94.5%, which is slightly higher than the method proposed in this paper. And after processing with the HEC method proposed in this paper, the identification result is 94.3%. The method proposed in this paper is higher than the method proposed in the SEARE and Sleepy studies and slightly lower than the method proposed in the RT-Fall study. As for the processing time, the processing time of Wi-CAS is significantly lower than that of several other systems. The comparison results prove that Wi-CAS can minimize the processing time while ensuring higher recognition results, and effectively improve the efficiency of the system.

#### 5.3.4. System Boundary Exploration

Home isolation does not always entail being isolated alone, as more often two or three people isolated at the same time. Therefore, when there are people doing other actions to interfere or multiple people doing actions at the same time, whether the system performance can be accurately identified is an important criterion. Accordingly, we set up four experimenters to verify two aspects of interference and simultaneous actions of multiple people. First of all, we investigate the influence of the number of interferers, let one experimenter do the action, and add one, two, three interferers respectively, the interferers do other actions in the room to interfere, and compare the recognition results with the results without interference, the comparison results are shown in Figure 18a. After that, 2, 3 and 4 experimenters were set up to perform the same action at the same time, and the results were compared with those of 1 person doing the action to compare all the data using the above classification method respectively. The experimental results are shown in Figure 18b.

As can be seen in Figure 18a, the action recognition result decreases when the number of interferers increases. When the number of interferers is 3, the recognition accuracy of Wi-CAS can still reach more than 60%, which proves that the Wi-CAS system has a good anti-interference property. Figure 18b shows that the action recognition of Wi-CAS decreases when the number of people rises. When two people do the action at the same time, the action recognition rate is still above 75%, when the number of people reaches three, the action recognition rate drops to 60.7%, and when four people do the action at the same time, the action recognition rate of the system drops to 44.3%, and the experimental results are as expected. However, when two and three people do the action at the same time, the system’s action recognition rate is still above 60%, and the experimental result Wi-CAS shows better performance within three people.

## 6. Conclusions

In this paper, we propose Wi-CAS, a non-contact indoor person continuous activity sensing method based on channel state information, which uses an ensemble classification model HEC based on hierarchical clustering for action recognition. Wi-CAS is divided into two phases: off-line and on-line. In the off-line phase, a large amount of daily action data is collected, and the texture of the action signal is extracted as features to construct the HEC model after filtering. In the on-line phase, real-time person activity data is collected, and the action is first separated using the DTW algorithm, after which the same processing is performed and input into the HEC model for the final recognition result. After extensive experimental evaluation, the average recognition rate of the Wi-CAS method reaches 94.3%, which proves that the method has high robustness as well as recognition accuracy.

We will continue to improve the performance of the system in future work and will aim to extend this approach to more areas.

## Figures and Tables

**Figure 1 sensors-21-08404-f001:**
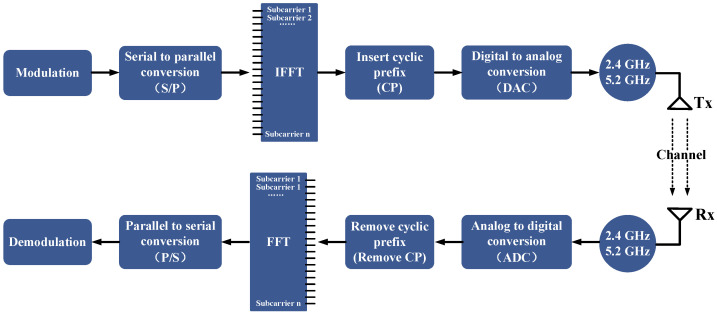
Principle of OFDM.

**Figure 2 sensors-21-08404-f002:**
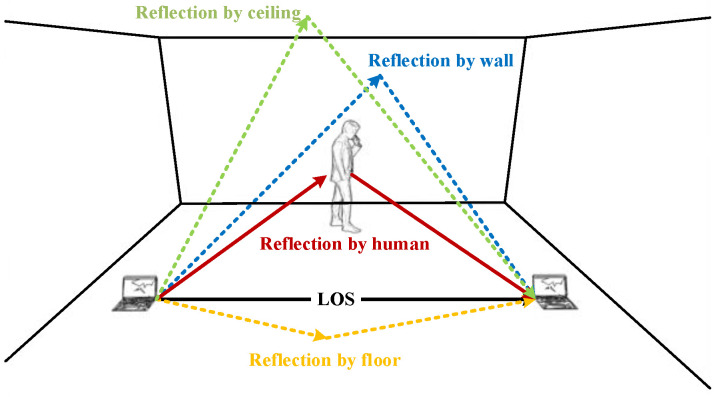
Wi-Fi sensing model.

**Figure 3 sensors-21-08404-f003:**
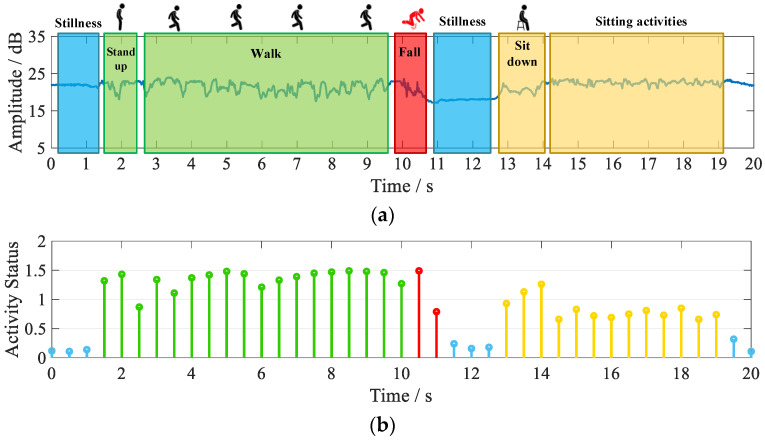
Human activity data: (**a**) human activity data; (**b**) human activity state diagram.

**Figure 4 sensors-21-08404-f004:**
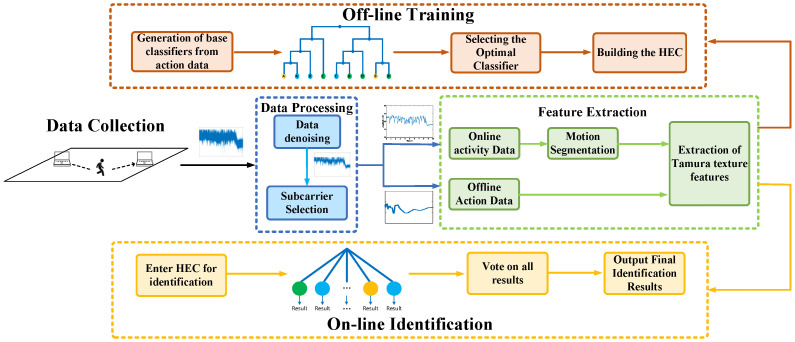
System architecture of Wi-CAS.

**Figure 5 sensors-21-08404-f005:**
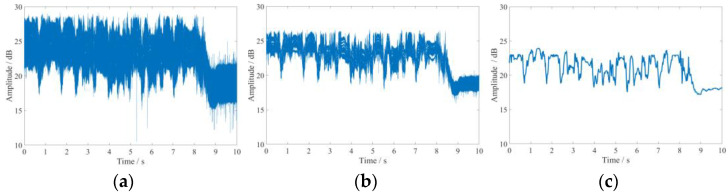
Motion data processing: (**a**) CSI raw motion data; (**b**) signal denoising; (**c**) subcarrier selection.

**Figure 6 sensors-21-08404-f006:**
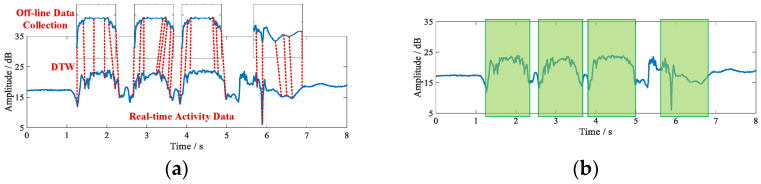
DTW algorithm process: (**a**) DTW matching process; (**b**) action matching results.

**Figure 7 sensors-21-08404-f007:**
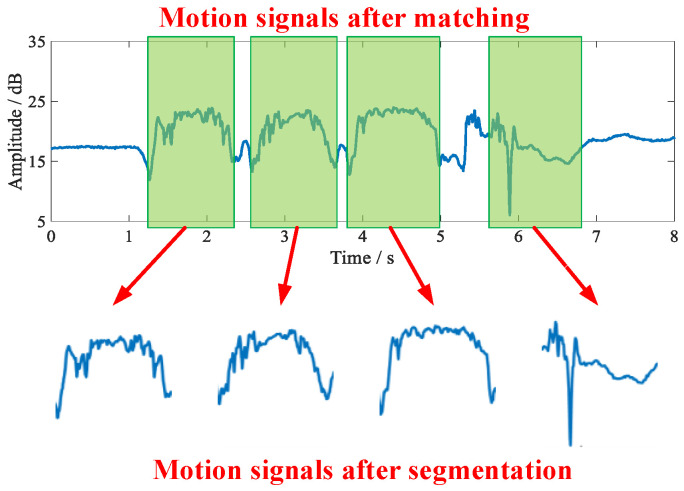
Motion signal segmentation.

**Figure 8 sensors-21-08404-f008:**
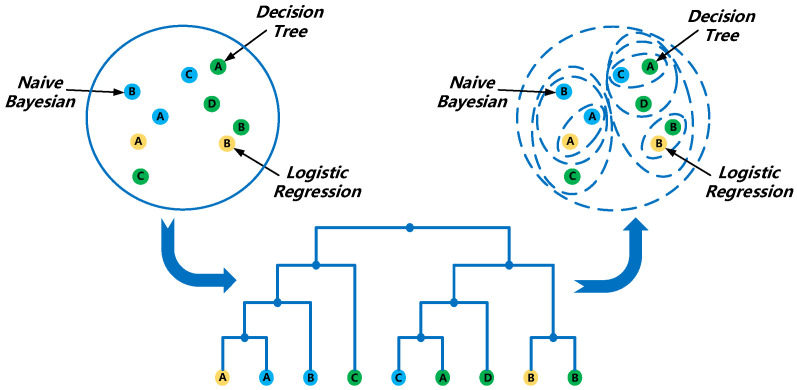
Hierarchical clustering process diagram.

**Figure 9 sensors-21-08404-f009:**
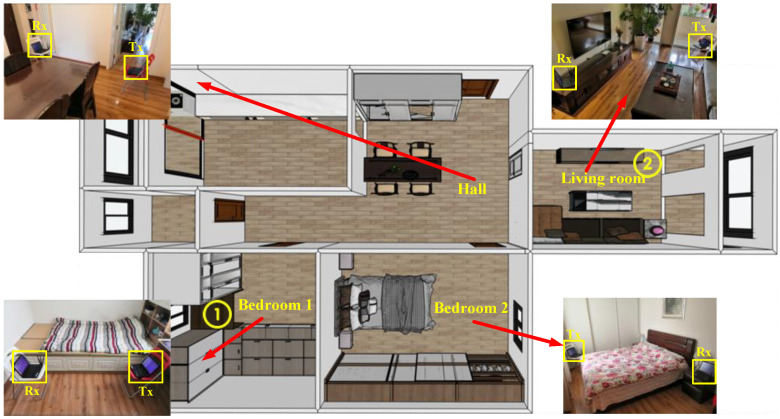
Experimental environment.

**Figure 10 sensors-21-08404-f010:**
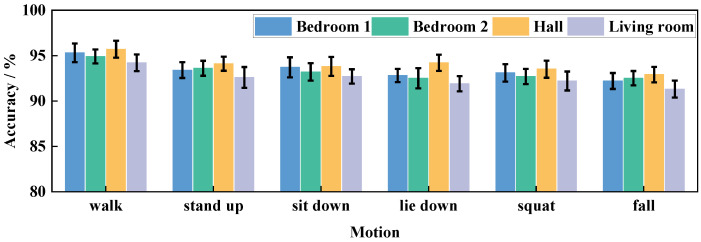
Cross-domain performance comparison.

**Figure 11 sensors-21-08404-f011:**
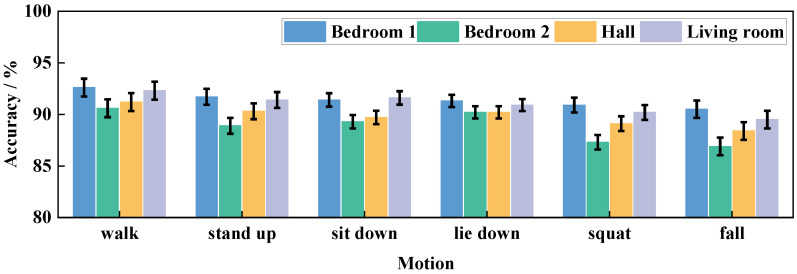
Through-wall performance comparison.

**Figure 12 sensors-21-08404-f012:**
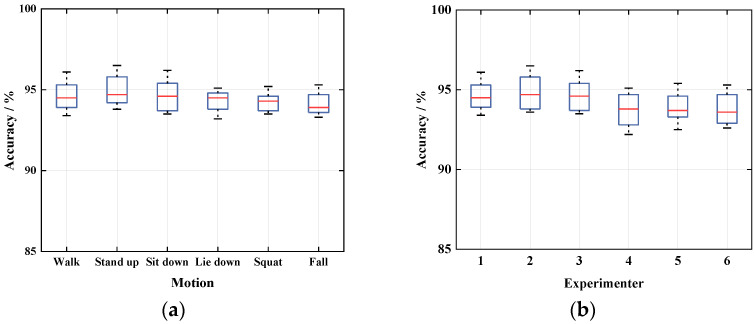
Impact of user diversity: (**a**) different movements of the same person; (**b**) the same action for different people.

**Figure 13 sensors-21-08404-f013:**
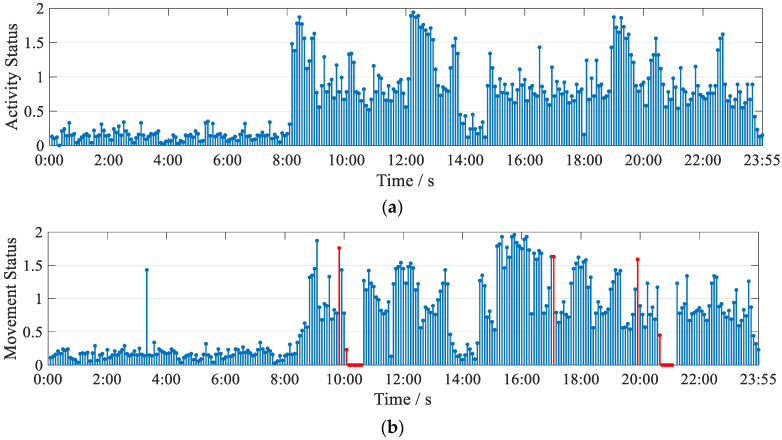
Activity status map: (**a**) experimenter 1 24 h activity status map; (**b**) experimenter 2 24 h activity status map.

**Figure 14 sensors-21-08404-f014:**
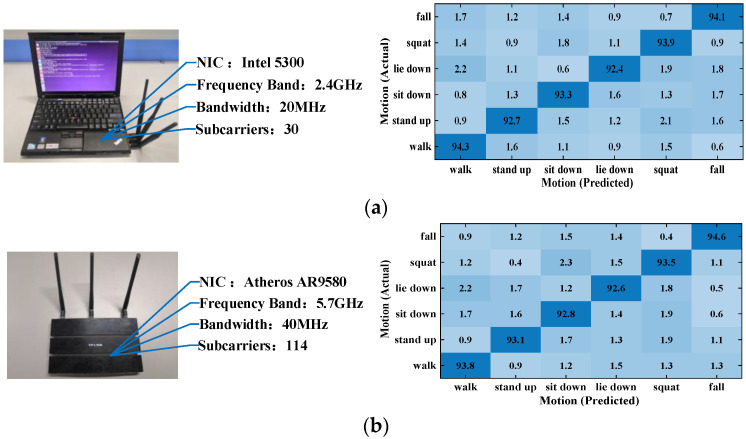
Cross-platform validation: (**a**) Intel 5300 NIC Platform; (**b**) Atheros AR9580 NIC Platform.

**Figure 15 sensors-21-08404-f015:**
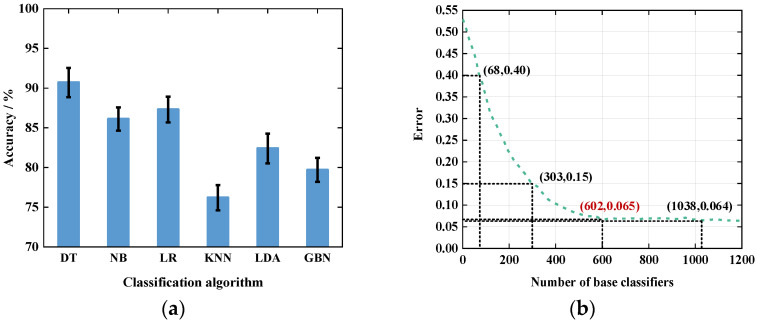
The impact of the type and number of base classifiers: (**a**) impact of the base classifier type; (**b**) impact of the number of base classifiers.

**Figure 16 sensors-21-08404-f016:**
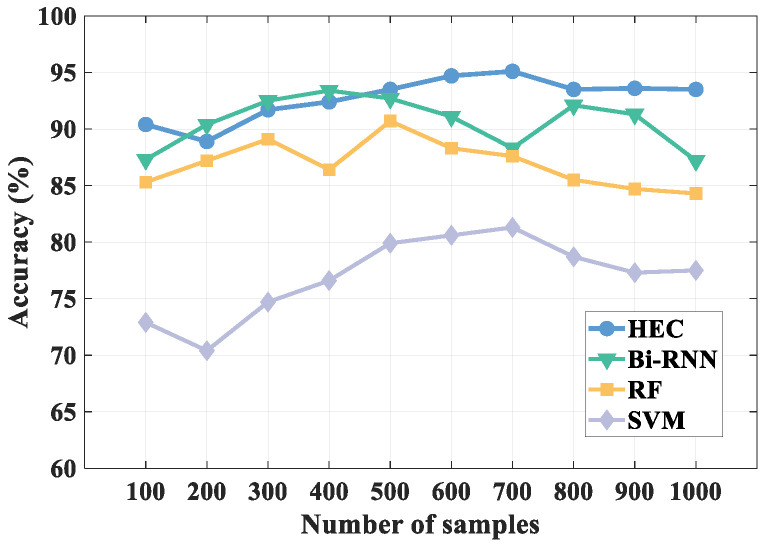
Impact of different sample sizes.

**Figure 17 sensors-21-08404-f017:**
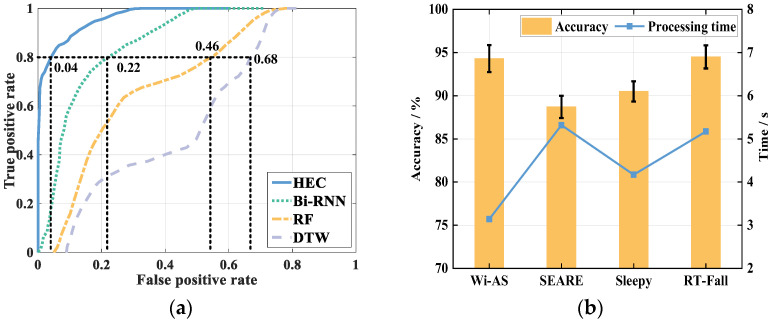
Comparison of different classification algorithms and systems: (**a**) comparison of different algorithms; (**b**) comparison of different systems.

**Figure 18 sensors-21-08404-f018:**
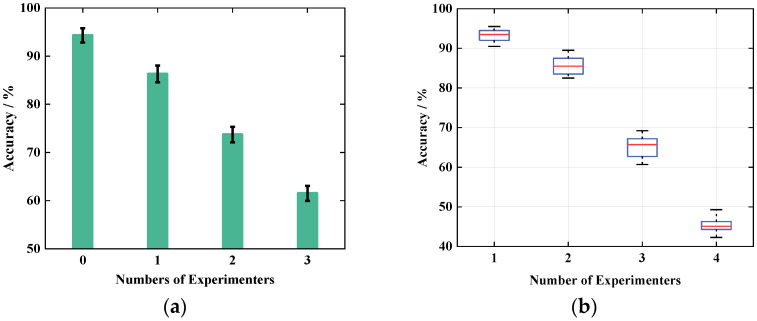
System boundary exploration: (**a**) impact of the number of interferers; (**b**) impact of multiplayer action.

**Table 1 sensors-21-08404-t001:** Experimental parameter setting.

Group	Gender	Age	Height	Movement Duration	Sample Size	Processing Time
Group 1	Male	10–28	1.20–1.80 m	10 s	30×20	3.12 s
Group 2	Male	29–60	1.70–1.90 m	10 s	30×20	3.23 s
Group 3	Female	10–28	1.20–1.65 m	10 s	30×20	2.96 s
Group 4	Female	29–60	1.60–1.75 m	10 s	30×20	3.34 s

**Table 2 sensors-21-08404-t002:** Twenty-four hour activity record of experimenter 1.

Time	Activity	Time	Activity	Time	Activity	Time	Activity
8:00–8:10	Get up and Washing	9:55–10:10	Room cleaning	14:45–15:45	Reading	20:25–22:20	Play Games
8:10–8:40	Fitness and Shower	10:10–12:00	Online class	15:45–17:55	Watch TV	22:20–22:35	Washing
8:40–9:10	Cook and Eat	12:00–13:35	Cooking and Eat	17:55–18:45	Online Meeting	22:35–23:35	Play Mobile
9:10–9:55	Online class	13:35–14:45	Nap	18:45–20:25	Cooking and Eat	23:35–8:00	Sleep

**Table 3 sensors-21-08404-t003:** Twenty-four hour activity record of experimenter 2.

Time	Activity	Time	Activity	Time	Activity	Time	Activity
8:45–8:55	Get up and Washing	9:55–10:30	Out the door	14:25–15:00	Play Mobile	19:10–20:30	Play computer (Fall)
8:55–9:25	Cook and Eat	10:30–11:30	Watching TV	15:00–17:00	Fitness and Shower (Fall)	20:30–21:05	Out the door
9:25–9:40	Play Mobile	11:30–13:20	Cooking and Eat	17:00–17:30	Watching TV	21:05–23:35	Play Mobile
9:40–9:55	Room Cleaning (Fall)	13:20–14:25	Nap	17:30–19:10	Cook and Eat	23:35–8:45	Sleep

## Data Availability

Data is contained within the article.

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
