# Peer review of "Wi-CAS: A Contactless Method for Continuous Indoor Human Activity Sensing Using Wi-Fi Devices"

_sensors, 2021, doi:10.3390/s21248404_

Round 1

Reviewer 1 Report

  1. The HEC full name appears in the abstract, and you should also add the full name when this term first appears in article.
  2. DTW first appeared in line78, but the full name appeared in line 337, dynamic time regularization (DTW) algorithm. In line646 has another full name, Dynamic Time Routine (DTW). The definition in line337 does not clarify the reference source, nor does it emphasize that this is the algorithm proposed by the author, and that the name of Table1 is Motion Signal Matching, which also does not specify that this is DTW, makes reading papers extremely difficult, and the definition in line646 makes such errors unacceptable.
  3. Line79, give Tamura texture feature the appropriate reference source and briefly describe how to apply the Tamura texture feature to the wi-fi signal feature. Sec. 4.3 explains the definition of the four characteristics of Tamura texture features, but it is not helpful for how to apply to Wi-Fi CSI signals. Especially Sbest(i,j) of eq(6) is the pixel position and mainly used in image processing.
    Line 363 lacks a sixth feature "roughness" and incomplete sentences.
  4. Please add the full name of any abbreviations used, even if they are well-known nouns (line169, FOD, FFT)
  5. Line 172, signals needs followed by a period.
  6. 3.1 The explanation about OFDM is very unprofessional. Line191 said that Figure 1 shows how to generate and transmit CSI in OFDM, but in fact Figure 1 is the transmission and reception diagram of OFDM. Line209 mentions that CSI is generated after IFFT, but the format and details are not mentioned at all, nor does it explain how CSI is used in this paper. In fact, CSI is not a set of signals generated and transmitted in the communication system. The usage here is wrong. The CSI mentioned in line233 and the CSI mentioned in line209 are two different concepts, and the concept of line233 is more correct. This may be a problem of English skills, the English of entire paper needs to be overhauled.
  7. Eq.2, Symbol ω is used incorrectly, x(t- ω), where ω is the number of channels? The notation of ω used in eq.2 is not proper. Or, the author uses the symbol of the continuous signal, but misuses the formula of the discrete signal. The author seems to be unfamiliar with the theory of communication systems. There are many statements in Sec. 3.1 and 3.2 that are unclear or have errors, such as Line 230, “x(t-ω) is the signal at the transmitter”.
  8. Eq. 4 There is no reference source. The descriptions of Sec 3.1 and 3.2 cannot help readers understand CSI, nor understand the usage of CSI in Wi-Fi systems. Compared with other papers, such as Z Tang et al. "Human Behavior Recognition Based on WiFi Channel State Information," in Proc. 2020 Chinese Automation Congress (CAC), the description of CSI is easier to read and easier to understand.
  9. Line 256, it is common in current Wi-Fi APs to have four to eight antennas, is it wrong for the authors to emphasize here that one antenna is used? The same question for use of three antennas at the receiver side, which means a 1x3 MIMO system is used in this study?
  10. As shown in Figure 3(b), Activity status is not defined. In line 270, it is mentioned that "… have more obvious differences in amplitude," where amplitude is directly equal to activity status so that readers cannot understand it. Generally speaking, the received intensity of Wi-Fi signal is between -25dBm and -100dBm, not between 0 and 2. And Figure 3 does not state the information about CSI, but rather like using RSSI for detection. The vertical axis unit Amplitude / dB in Figure 3(a) should also be clearly stated, otherwise it is a definition that does not conform to mathematical logic. Or the author wants to express Amplitude (dB) for raw CSI.

Based on the above comments, it is unclear in the basic narrative of this paper, and there are many errors in English. Section 3 Preliminaries does not provide sufficient clear and correct background knowledge. The content of Sec 4.3 and 4.4 is also superficial and has no direct link with the proposed system. Authors should re-examine their papers and try to resubmit them.

Reviewer 2 Report

The paper proposes a non-contact indoor person activity sensing method based on Wi-Fi signals. The work is interesting and clearly written, the methodological and experimental parts are both described in an extensive and exhaustive way. The work is rich in details and could be more concise, especially in the part relating to algorithms, but this detailed discussion may be useful to readers interested in replicating these techniques.

Just a few small observations:

  • all acronyms should be clarified at their first use (e.g. DTW).
  • there are some typos in the text (e.g. after the dot there are words starting with a lowercase letter)
  • In the confusion matrices of Fig. 13, the horizontal and vertical axes are both named with "Motion". But which of the two refers to the real classes and which to the predicted classes?

Round 2

Reviewer 1 Report

In this revision, the author has made sufficient revisions and supplements based on the review comments I put forward, and all the questions raised have good responses. However, there are too many mistakes and ambiguities in the writing ability of this paper. In the last review comments, I only made 10 suggestions, and did not write out all the problems that I saw. I hope the author can see a straw shows which way the wind blows. Review the paper again to improve the readability of the paper.

It is a pity that the author has not adjusted the other parts except for the 10 suggestions I made, and the readability of the whole paper is still insufficient. I would like to cite a few examples for reference:

1. Writing:

(a). Line 182, "(OFDM) is a technique used to reduce the effects of channel frequency and time selectivity," should be “(OFDM) is a technique used to reduce the effects of frequency selectivity and time selectivity," or clearer “(OFDM) is a technique used to reduce the effects of frequency-selective and time-selective fading,"

(b) Line 352, “So only DTW is used for action segmentation in this paper.” Should be “So DTW is only used for action segmentation in this paper.”

2. Readability:

(a) Line 345, DTW still did not provide a reference source, or did not provide a detailed approach, so in Algorithm 1, 1: dist = dtw(t, r), it is no way to know how to calculate, and t and r are undefined.

(b) The description of Algorithm 1 is vague, lacks a clear definition of symbols, and lacks calculation methods (such as dtw(t,r), t(i,:), r(i,:)). Line 359, "Assuming that the real-time activity signal collected Off-line is set to M" is unclear. What is M? A set or a scalar? Is a vector on a time series, or other? Line 360, 361, "M = x1, x2, …, xm, and M are m". What a strange way to define M! So again, what is M? These instructions do not carry any useful information, plus the problem that mentioned above symbols are not defined, the algorithm is fundamentally unreadable. There are the same problems in the second half of the paper, I will not state them one by one.

Others:

(a) eq. 4, H(f) should be H(fn)

Round 3

Reviewer 1 Report

This paper proposes a behavior detection system based on Wi-Fi signals, which is used in the safety monitoring system of COVID-19 home isolation, which is urgently needed. The topic meets the current needs, and the author has conducted a lot of experiments to prove the feasibility of the system. The system uses the DTW algorithm to segment the real-time signal, then uses Tamura texture features for feature marking, and finally uses the HEC algorithm to classify the segmented real-time signal to determine the user's current state. The biggest problem with this paper right now is that it's not written well enough to make it impossible for reviewer to agree intuitively with the results of their experiment.

Let's go back to Sec. 4.2 Motion Segmentation. At the first review, the author mistook the full name of the DTW. At the second review, the author did not give a reference to DTW, which would have led the reader to misunderstand that algorithm 1 was the author's new approach (although it had the same name as a well-known algorithm, DTW). As mentioned in the first review, "the definition of line 337 does not clarify the reference source, nor does it emphasize that this is the algorithm proposed by the author, and that the name of Table1 is Motion Signal Matching, which also does not specify that this is DTW, makes reading papers extremely difficult". In the third revise, a new reference was given to line 350. However, the reviewer went to reference this document, it didn't mention DTW, and there was no clear method of segmentation. The author did not explicitly answer the question I mentioned in the second review, and did not answer what is dtw(t,r), and what are t and r? In fact, when the authors wrote Sec3 and Sec4, they forwarded the contents of other literature directly to their paper, but did not integrate them with their own system, such as Sec 3.1 and Sec 4.3 mentioned in the first review, they are separate fragments. Reviewer read very hard these contents but found that the content has nothing to do with the system, the author filled in a bunch of mathematical formulas and algorithms but ultimately did not use in the system. In fact, these contents can be omitted in a streamlined manner, helping the reader to understand more easily whether the proposed system is reasonable and feasible.

Let's go back to algorithm 1 again, after reviewer verification, in fact, it is the DTW algorithm, the author transferred it from other literature, but still did not integrate it with their own system. In other literature you can find a sentence like "Let t and r be two vectors of lengths m and n, respectively. The goal of DTW is to find a mapping path such that the distance on this mapping path is minimized", where t and r is used by the author in algorithm 1 directly. But to integrate with the proposed system, it should be changed to M and N. Instead of changing t and r to M and N, the authors wrote them all together to make the algorithm more confusing. Let the viewer feel that the author also does not understand DTW, stuffing things used to lengthen the length of the paper. DTW has ready-made packages available in Maltab and Python, eliminating the need to write the algorithm over and over again in the paper. At the same time, the algorithm 1 worked out the shortest distance D(m, n), and did not explain the method of the motion segmentation, the section content is seriously inconsistent with the title.
